# Logistics and Supply Chain Resilience of Japanese Companies: Perspectives from Impacts of the COVID-19 Pandemic

**Rajali Maharjan [1,\*] and Hironori Kato [2]**

[1]  Japan Transport and Tourism Research Institute, Tokyo 105-0001, Japan
[2]  Graduate School of Engineering, The University of Tokyo, Tokyo 113-8656, Japan
\*  Correspondence: maharjan-reu@jttri.or.jp

**Abstract:** *Background:* Enhancing the resilience of global supply chains has become of increasing priority in response to recent natural disasters and the COVID-19 pandemic. *Methods:* This paper presents findings from semi-structured interviews with five Japanese companies conducted between November 2020 and February 2021 to examine the impacts of the COVID-19 pandemic on different aspects of logistics and supply chain activities and resilience strategies implemented. The interviews focused on firms' financial performance and the status of preparedness, response, and future plans from the perspective of logistics and supply chain resilience. Through interviews, we also investigated whether existing logistics and supply chain resilience strategies helped the companies avoid, withstand, respond to, or recover from the pandemic's impacts. *Results:* The results indicated that the interviewed companies experienced both positive and negative impacts from the pandemic on their logistics and supply chain activities and experienced negative impacts mainly on their financial performance. *Conclusions:* A certain level of preparedness was observed; however, the levels of resilience preparedness, response, and future plans varied among companies with different attributes, such as industry type and organisation size.

**Keywords:** supply chain resilience; logistics resilience; COVID-19 pandemic; Japanese companies; firm performance

## 1. Introduction

The COVID-19 pandemic has placed unprecedented stress on logistics and supply chain activities and presented many urgent challenges for industries worldwide. The pipelines of global supply chains, which span from the supply of raw materials to the delivery of products, were heavily affected by the pandemic, and disruptions were observed in all supply chain phases [1]. International logistics for maritime, air, and terrestrial routes experienced delays, postponements, cancelations, and obstructions due to large-scale travel restrictions and the closing of borders [1]. Bottlenecks in transportation and logistics disrupted the movement of materials and products along the supply chain. Furthermore, the pandemic caused disturbances in supply and demand at both global and local scales [2]. In a survey of 1000 companies in different parts of the world conducted by the Capgemini Research Institute from August to September 2020, >80% of the organisations reported that their supply chains were negatively impacted by the COVID-19 pandemic, with a vast majority struggling across all aspects of their operations [3]. The pandemic has forced organisations to prioritise supply chain resilience, with two-thirds stating that their supply chain strategy will need to change significantly to adapt to the 'new normal'. As such, many organisations are now realising the strategic importance of resilience investments.

Many studies have indicated that supply chain resilience, or the ability of a supply chain to respond quickly to disruptions, is one of its most important challenges [4–9]. Given the widespread impact of the COVID-19 pandemic, the adoption of logistics and supply chain resilience strategies is critical for ensuring that companies can avoid, respond to,

and recover from disruptions. It has been pointed out [1] that enhancing supply chain resilience is the key driver for reducing vulnerability during disruptive times. Thus, there is a growing need for companies to build resilient supply chains [9].

The resilience of logistics and supply chains can be improved via the implementation of strategies that target both the nodes and links of the supply chain, which we call logistics and supply chain resilience strategies (SCRESTs). Figure 1 illustrates the typical structure of a supply chain network. A supply chain consists of nodes (i.e., suppliers, manufacturing/production centres, and facilities such as logistics centres, distribution centres, warehouses, and customers) connected to different links. When a node or link in the network is disrupted by even a small event, it can have major consequences for the entire network. A failure in a node or link can potentially stop the flow of materials across the network. Disruptions arising from natural disasters, manmade disasters, pandemics, epidemics, government regulations, etc., can affect the nodes and/or links of the logistics and supply chain network, necessitating the implementation of appropriate resilience strategies.

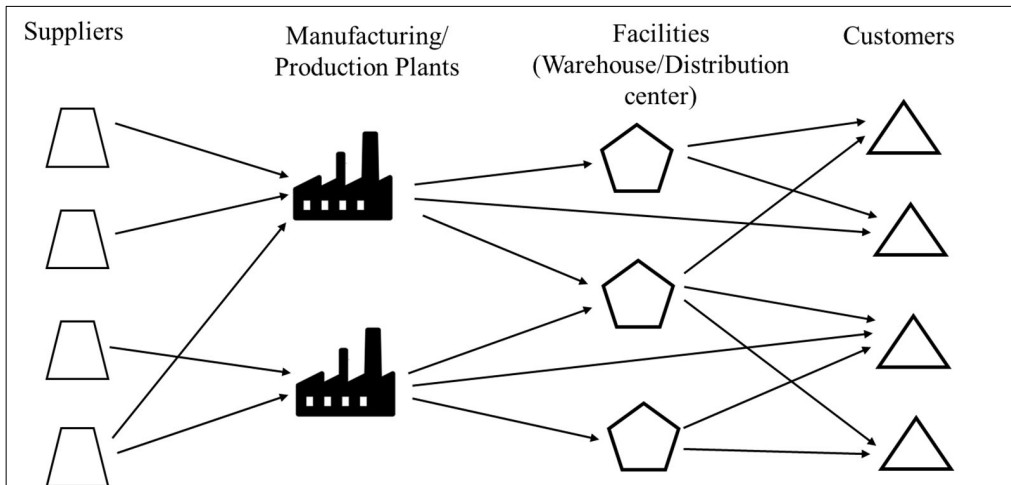

**Figure 1.** Typical structure of a supply chain network.

Resilience strategies are adopted to avoid, withstand, and recover from disruptions in supply chain networks. Supply chain resilience strategies are important because they enable efficient and effective responses [10]. For enhancing the resilience of supply chains, researchers have proposed strategies such as redundancy in inventory, additional production capacity, diversification of suppliers, and making supply chains shorter, more domestic, and more diversified [1,10,11]. However, Miroudot [11] argues that before redesigning global supply chains, it is necessary to identify the problems faced by firms during crises and the policies that can resolve them. Miroudot [11] also reported that many proposed solutions tend to be disconnected from the conclusions of supply chain literature (e.g., reshoring does not lead to resilience) and insisted that the insights of international business and global value chain researchers can contribute to the development of better solutions.

In this context, the objectives of this study were to (1) analyse the impacts of the COVID-19 pandemic on logistics and supply chain activities and firms' financial performance through a case study of five Japanese companies; (2) identify the current status of resilience preparedness, response, and future plans from the logistics and supply chain perspective for the different Japanese companies; and (3) determine whether the existing logistics and supply chain resilience measures implemented by the interviewed companies helped to avoid, withstand, respond to, or recover from the impacts of the COVID-19 pandemic. To the best of our knowledge, no empirical study thus far has addressed all the topics examined in this study.

The remainder of this paper is organised as follows. Section 2 presents a literature review. Section 3 describes the research methodology. The results are reported and dis-

cussed in Sections 4 and 5, respectively. Section 6 explains the policy implications. Finally, Section 5 concludes the paper.

## 2. Literature Review

In this section, we summarise previous studies on the impacts of COVID-19 on the logistics and supply chain activities of Japanese companies, the status of SCRESTs implementation, and the benefits associated with its implementation.

### 2.1. Impacts of the COVID-19 Pandemic on Japanese Companies

Among the diverse impacts of the COVID-19 pandemic on the Japanese logistics sector are disruptions in the maritime, air, and land transportation sectors, which have been affected mainly by measures for preventing the spread of the virus. For instance, maritime transportation has been affected by the suspension of services in some ports, air transportation by the decline in the number of passenger air service operations, and land transportation by the blockading of roads and railway tracks [12]. These virus-spread prevention measures had a significant negative impact on economic activity and was able to severely affect the quality of life in Japan. It should be noted that maritime transportation accounts for 99.7% of Japan's international trade volume, air transportation for 40% of the trade revenue, and land transportation and warehouses combined for nearly 80% of Japan's overall logistics costs. Moreover, owing to differences in the features and functionalities of these transportation modes, there is a limitation on the substitutability of each logistics route [12].

Regarding the impact of the COVID-19 pandemic on logistics facilities, CBRE's Japan Logistics Occupier Survey asked respondents (firms using logistics facilities in Japan) a series of questions to gauge the short-, medium-, and long-term impacts of the COVID-19 pandemic on logistics demand in March 2020 [13]. Short-term impacts such as shortages of warehouses and delivery workers were reported, and 405 respondents reported increases in cargo and delivery volumes. Regarding the medium- to long-term impacts of the pandemic on businesses, 30% of respondents from 361 companies mentioned additional inventory for unexpected situations, and 17% mentioned accelerated automation of warehouse operations. Additionally, a survey conducted by Japan's Chamber of Commerce and JETRO's overseas offices in China, Malaysia, Indonesia, India, and the USA from April to July 2020 revealed that Japanese companies overseas observed operational decline mainly because of reductions in domestic and overseas demand, operational regulations by the government, disruptions to domestic and overseas supply chains, logistical constraints, and increased costs [14]. The findings suggested that reactive resilience measures alone are insufficient to prevent or deal with the consequences of disruptions within and outside of Japan.

Ino and Watanabe [15] investigated the impacts of the pandemic on manufacturing industries in Japan from a global supply chain perspective and the impact of the decentralisation strategy promoted by the government of Japan. They examined whether the overconcentration of manufacturing bases would improve the resilience of Japanese companies' global supply chains. Ito [16] investigated the impact of supply chain disruptions caused by the pandemic on manufacturing in Japan. Using monthly production data, monthly export and import data, Japan's input–output tables, and international input–output tables, researchers found that manufacturing activities in Japan were negatively impacted by supply chain disruptions due to COVID-19. Zhang [17] investigated the impacts of COVID-19 and policy responses on global production and supply chains using aggregate-level data for Japanese multinational corporations (MNCs). The results indicated that COVID-19 had significant negative impacts on the global supply chains, firm performance, expectations, and business plans of Japanese MNCs in major host countries.

### 2.2. Implementation of SCRESTs by Japanese Companies

Studies have highlighted the importance of implementing strategies to improve the resilience of logistics and supply chains activities [18,19]. Kiers et al. [20] highlighted the

importance of implementing specific strategic changes to reduce the impact of disruptions and shift the focus from cost efficiency to supply chain resilience.

### 2.3. Benefits of Implementing Logistics and Supply Chain Resilience Strategies

Resource allocation (i.e., investment) and the implementation of appropriate SCRESTs are needed to improve the resilience of supply chains. However, while the implementation of SCRESTs is desirable and critical for withstanding disruptions, a question often asked regarding resilience is, 'Are the investments worth it?' Resilience enhancement often requires large investments, which should be justified by substantial projected returns; however, monetising the paybacks is difficult [21]. Therefore, it is important to investigate the financial benefits of supply chain resilience to justify resource allocation (e.g., investments) [22].

Wieland and Wallenburg [23] revealed that improved supply chain resilience in terms of agility and robustness enhances a supply chain's customer value (i.e., customer satisfaction). Govindan et al. [24] found that the resilient practice of flexible transportation was positively related to customer satisfaction. The findings of an empirical study performed by Li et al. [25], in which supply chain preparedness was directly linked to firms' financial performance, support the assertion that prepared supply chains enjoy better financial outcomes. Additionally, they suggest that supply chain preparedness practices tend to be more financially beneficial to a firm than reactive practices. The literature also indicates that prepared supply chains experience fewer negative stock-market reactions in the face of disruptions [26].

### 2.4. Research Gaps

Supply chain resilience is a relatively new management concept; therefore, empirical research on the link between supply chain resilience and performance is limited [25]. Chowdhury et al. [27] systematically reviewed COVID-19-related supply chain studies and found that only 6 of the 73 studies reviewed used empirical methods, highlighting a general lack of empirical studies. The gaps in the existing literature are as follows. First, although the implementation of SCRESTs is important for dealing with future disruptions, there have been no studies on the SCRESTs implementation statuses of companies in different industry sectors. Second, there is limited empirical evidence on whether companies' logistics and supply chain preparedness activities to enhance resilience have helped them avoid, withstand, respond to, or recover from the impacts of the COVID-19 pandemic. Third, the impact of the COVID-19 pandemic on Japanese companies was investigated using secondary data in only three of the studies. Moreover, no attempt has been made to investigate the impacts of the pandemic on the logistics and supply chain activities of Japanese companies in different sectors.

## 3. Research Method

In this study, a qualitative approach was used to obtain the necessary information. We used an exploratory research design and performed no statistical analyses. An interview-based approach, which is the most common format of data collection in qualitative research, was selected because it is an excellent way to gather detailed information, and the respondents' opinions are representative of the scope of matter under study [28]. We employed a combination of purposive and snowball sampling techniques to interview key informants from companies in global logistics and supply chain networks. A combination of purposive and snowball sampling techniques allowed maximum variation, following the principles of appropriateness and adequacy [29,30]. The main criterion for selecting respondent companies was whether they engaged in global logistics and supply chain activities.

Semi-structured questionnaires were used in the interviews, as the primary purpose was to identify as many important problems, concerns, and discussion points relevant to the research field as possible [31]. The interview questions were designed using the funnelling technique [28,31]. The questions ranged from broad topics related to the company's logistics

strategy and supply chain activities, to specific topics, including questions on the impact of the pandemic on logistics and supply chain activities; the financial performance of firms; the status of logistics and supply chain resilience; the relationship between logistics and supply chain resilience activities; and the ability to avoid, withstand, respond to, and recover from the impacts of the COVID-19 pandemic. The interview questions were broadly classified as follows:

1.  What is the impact of the COVID-19 pandemic on logistics and supply chain activities?
2.  What is the impact of the COVID-19 pandemic on the firm's financial performance?
3.  What is the status (past, present, or future) of logistics and supply chain resilience?
4.  Did existing logistics and supply chain resilience measures help the firm withstand, respond to, or recover from the impacts of the COVID-19 pandemic?

The interviews were conducted in Japanese using the online platforms Zoom and Microsoft Teams between October 2020 and February 2021. The respondents were at the decision-making level and included both directors and general department managers. In total, five in-depth interviews were conducted with the interviewees, and each interview lasted approximately 1 h on average. All conversations were recorded with permission from the respondents to ensure that correct information was gathered from the interviews. Table 1 presents brief profiles of the companies interviewed.

**Table 1.** Respondent company profiles.

| Company | Industry | Size | Customer Base | Interview Date | Interviewee(s) |
|---|---|---|---|---|---|
| Company 1 | Cosmetic | Large enterprise | Supermarkets; wholesalers; retailers. | 22 October 2020 | Director of the logistics centre |
| Company 2 | Trading | Small and medium enterprise | Manufacturing companies; assembly companies; maintenance companies. | 1 November 2020 | Sales head |
| Company 3 | Logistics and supply chain solutions | Large enterprise | Manufacturing companies; assembly companies; production companies; trading companies. | 10 November 2020 | General managers |
| Company 4 | Freight forwarding | Small and medium enterprise | Manufacturing companies; agricultural machinery parts companies; pharmaceutical companies; apparel companies; convenience goods companies. | 22 December 2020 | Executive director |
| Company 5 | Freight forwarding | Small and medium enterprise | Apparel business companies; trading companies. | 17 February 2021 | Director |

## 4. Findings

### 4.1. Impacts of the COVID-19 Pandemic on Logistics and Supply Chain Activities

The COVID-19 pandemic has caused considerable damage to different industry sectors, with varying impacts on different dimensions. In this study, we examined the impacts of the pandemic on the logistics and supply chain activities of five Japanese companies, as summarised in Table 2. These impacts, based on the interviewees' responses, are presented in relation to the structure of logistics and supply chain networks. We observe that different companies faced different types of impacts. Company 1 experienced an increase in demand for its products, which was a positive impact, whereas Company 2 experienced a reduction in demand for its products and/or services and difficulty in accessing its suppliers, which were negative impacts. Companies 3 and 4 experienced both positive and negative impacts, as they saw increases in the demand for their products and/or services (positive impact) and subsequent increases in sea and air transportation costs (negative impact). Company 5 primarily experienced negative impacts owing to high

transportation costs, reduced shipping volumes, and irregular shipping schedules. We also note that Company 1 experienced impacts mainly on the nodes of the supply chain network, whereas Companies 2–5 experienced impacts on both the nodes and links.

**Table 2.** Impacts of the COVID-19 pandemic on logistics and supply chain activities and firms' financial performance.

| Company | Impacts on | |
|---|---|---|
| | **Logistics and Supply Chain Activities** | **Firm's Financial Performance** |
| Company 1 | Increase in demand for sanitary products. | N/A |
| Company 2 | Decrease in demand from customers; No access to suppliers from March to August. | Difficult to determine yet, should wait until the end of March |
| Company 3 | Increase in demand for warehouse storage; High air transportation costs; High sea transportation costs. | Neither loss nor gain |
| Company 4 | Increase in demand for warehouse storage; High air transportation cost; High sea transportation cost. | Negative impact; exact details will be known at the end of the fiscal year |
| Company 5 | High air transportation cost; High sea transportation cost; Decrease in shipping volume; Irregular shipping schedule. | Negative impact; sales volume decreased by 20–30% |

### 4.2. Impact of the COVID-19 Pandemic on Firms' Financial Performance

As shown in Table 2, Company 1 did not comment on the impact of COVID-19 on the firm's financial performance. Company 2 stated that the impact of COVID-19 was not yet clear, given that their financial performance is generally evaluated only once a year. Company 3 did not observe a significant improvement or decline in its financial performance. Company 4 experienced a negative impact; however, the exact details would be known only at the end of the fiscal year (March 2021). Company 5 experienced a negative impact on its financial performance, mainly because of a reduced sales volume. These responses highlight the importance of correctly timing the interviews to obtain necessary information on the impact of COVID-19 on the financial performance of Japanese companies.

### 4.3. Status of SCREST Implementation

The interviewed companies' SCRESTs were evaluated according to their preparedness, responses, and intended future initiatives. Table 3 presents the logistics and supply chain resilience preparedness, responses, and intended future initiatives of the interviewed companies. For preparedness, we asked about initiatives undertaken for natural disasters and pandemics, focusing on the COVID-19 pandemic, and future initiatives intended for both natural disasters and pandemics.

With regard to SCRESTs for preparedness, Company 1 implemented strategies such as business continuity planning, lateral transhipment between logistics centres, and moving electrical lines away from tsunami-prone areas as part of resilience preparedness. The respondent from Company 1 pointed out that all SCRESTs were implemented after the 2011 Great East Japan Earthquake. Companies 2 and 4 did not show preparedness. Company 3 had facility fortification and dispersion as part of their SCRESTs for preparedness. Company 5 had prearrangements for using multiple ports (both sea and air) to handle incoming and outgoing cargo. This strategy protects the company from the risk arising from the unavailability of ports, allowing them to use an alternative port. The interviews revealed that none of the companies had strategies in place for dealing with pandemics. Companies 1, 3, and 5 had some form of SCRESTs in preparation for natural disasters, while Companies 2 and 4 did not have any SCRESTs for preparedness. From these results,

we conclude that larger firms tend to have more preparatory initiatives, which agrees with the results of previous studies [32,33].

With regard to SCRESTs for responding to the pandemic, our interviews revealed that Company 1 employed measures such as changes in transportation modes and the lateral transhipment of goods from low- to high-demand logistics centres. Company 2 implemented measures such as seeking alternative suppliers for products with a single supplier. Company 3 employed measures such as using other companies' warehouses in remote areas, moving stocks to overseas warehouses, and operating temporary warehouses overseas. Company 4 selected an alternative mode of transportation to respond to the COVID-19 pandemic, whereas Company 5′s response strategy was to wait for the situation to return to normal.

**Table 3.** Status of logistics and supply chain resilience.

| Company | Preparedness | | Response | Future Plans |
| | Natural Disaster | Pandemic | | |
|---|---|---|---|---|
| Company 1 | Business continuity plan; Provision of lateral transhipment between logistics centres; Moving electrical lines away from tsunami-prone areas. | None | Change of transportation mode; Lateral transhipment of goods from low-demand logistics centres to high-demand logistics centres. | Research on accurately predicting demand, multiple sourcing, increasing the capacity of logistics centres, and increasing the inventory of raw materials in Japan. |
| Company 2 | None | None | Seeking alternative suppliers for products with a single supplier. | Diversification of business. |
| Company 3 | Facility fortification; Facility dispersion. | None | Using warehouses of other companies; Moving stock to overseas warehouses; Operating a temporary warehouse overseas. | No actions have been planned yet. |
| Company 4 | None | None | Seeking alternative modes of transportation. | No actions have been planned yet. |
| Company 5 | Using multiple ports. | None | Wait for the situation to improve. | Enhance business relationship. |

The response strategies selected by the companies suggest that the selection of strategies is affected primarily by the nature of the impact that the companies experience, and the industry. For example, Company 1—a cosmetics company that experienced a surge in demand for sanitary products—chose to change the mode of transportation for importing the necessary goods from maritime to air to minimise the transportation time and meet the demand on time. Company 3—a logistics and supply chain service provider that observed a surge in demand for warehousing space—flexibly adopted strategies such as collaboration with other companies to obtain more warehouse space and moved finished goods to overseas warehouses to meet the demand. They faced several challenges in implementing their response strategies owing to labour shortages, sealed borders, trade control, and the shutdown of commercial aviation. In contrast, Company 5, which is in the freight-forwarding business, chose to wait for the situation to improve.

When the respondents were asked about their companies' plans to enhance resilience, their answers differed. For example, an interviewee from Company 1 mentioned that the firm would reduce uncertainties in future demand by introducing strategies such as increasing the capacity of the logistics centre and expanding the inventory of raw materials in Japan to prevent shortages. A respondent from Company 2 mentioned business diversification. Interviewees from Companies 3 and 4 indicated that their firms had no plans regarding future actions. However, a respondent from Company 5 mentioned that for small companies, it is crucial to have good business relationships with other companies;

therefore, it will focus on enhancing business relationships with other companies in the future. These findings imply that there is no coherence in future plans to enhance the resilience of logistics and supply chain activities among the companies interviewed. This could be because the uncertainties associated with the timing and scale of the crisis depend on the types of disruptions and the characteristics of the industry.

### *4.4. Benefits of SCREST Implementation*

To gather information on whether the companies with resilience preparedness thought they benefited from their preparedness initiatives for natural disasters against the impacts of the COVID-19 pandemic, respondents were asked whether they thought resilience preparedness helped their companies avoid, withstand, respond to, or recover from the impacts of the pandemic. Company 1 thought their resilience preparedness was helpful, whereas Companies 3 and 5 thought theirs were not. This question did not apply to Companies 2 and 4, because they did not have resilience preparedness strategies in place.

To examine why the responses of Companies 1, 3, and 5 differed, we compared their resilience strategies to the impacts observed. Lateral transhipment was implemented by Company 1, implying that there was a prior agreement and mechanism in place to move goods between logistics centres in the same echelon when needed. Company 1, whose main product was cosmetics, viewed the large surge in demand for sanitary products as an impact of the COVID-19 pandemic. Using their lateral transhipment strategy, this company moved goods from low-demand logistics centres to high-demand logistics centres to meet the increased demand, preventing sales losses.

In contrast, Company 3 implemented facility fortification, which refers to the retrofitting of the facility structure, and facility dispersion, which refers to having facilities in more than one location, as resilience strategies. This company, whose main business is providing logistics and supply chain solutions, viewed the large increase in demand for warehouse spaces and increased transportation costs as impacts of the COVID-19 pandemic. Neither of the resilience strategies implemented by Company 3 provided additional space on the scale that the company needed. Company 5, which implemented a strategy of using multiple ports within Japan, did not find its preparedness strategy useful because of the global and large-scale impacts of the COVID-19 pandemic affecting all ports equally.

## 5. Discussion

Companies in different sectors of Japan have faced significant challenges owing to the pandemic. The impacts of the pandemic were mixed; while some companies experienced positive impacts, others observed negative impacts. The impacts varied mainly by industry sector and company size. In addition to fluctuations in supply and demand, the main factors responsible for the operational decline of Japanese companies during the pandemic included disruptions to domestic and overseas supply chains, logistical constraints, high costs, and irregular shipping schedules.

Regarding the implementation status of SCRESTs, we found that three of the five companies we interviewed had some level of preparedness for logistics and supply chain resilience, specifically in anticipation of natural disasters. As part of their preparedness for disruptions, the interviewed companies had SCRESTs such as business continuity plans, lateral transhipment, facility fortification, facility dispersion, and the use of multiple ports. Two large enterprises and one small and medium-sized enterprise had SCRESTs for natural disasters. Furthermore, we found that whether companies implemented SCRESTs or not depended on the respective industry sector. Trading and freight forwarding companies tended to not have SCRESTs, which is attributed to these companies not having to take responsibility for the impacts of disruptions, because they were only service providers. Additionally, we found that companies that had experienced negative impacts from past natural disasters tended to implement SCRESTs.

Regarding the benefits of SCREST implementation, among the three companies that had implemented SCRESTs before the pandemic, only Company 1 observed benefits from

their preparedness for natural disasters, while Companies 3 and 5 did not. From Company 1's plans for implementing SCRESTs, we can infer that companies that have previously observed benefits are more willing to implement SCRESTs.

Based on the interview findings, we formulated the following hypothesis, as illustrated in Figure 2: the implementation of SCRESTs is influenced by the company size, industry sector, past disaster experience, COVID-19 impacts, and the benefits of implementing SCRESTs.

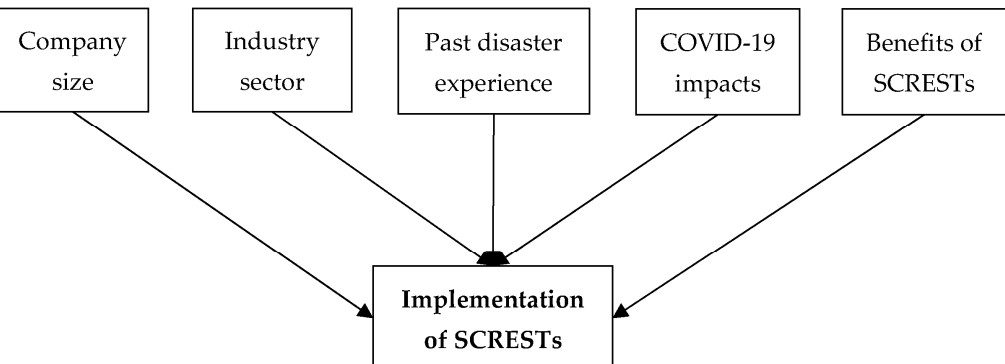

**Figure 2.** Factors influencing the implementation of SCRESTs.

## 6. Policy Implications

In response to the disruptions caused by the pandemic, the Japanese government implemented several initiatives to enhance the resilience of Japanese companies. These include providing subsidies for implementing strategies [34,35], promoting investment to strengthen supply chains [36], and investing in supply chain resilience in the Indo-Pacific region [37,38]. Providing subsidies to implement resilience strategies was the most relevant initiative to this study. The Japanese government provided earmarked funding and subsidies for companies to shift their production bases to Japan or countries other than China. While this initiative is commendable, the interview results suggested that it is irrelevant to the studied companies, because the strategies they used to respond to the impacts of the pandemic differ from those for which subsidies were provided. Therefore, none of the companies interviewed benefited from government initiatives, nor did the policy initiatives affect the interviewed firms' future resilience strategies.

Therefore, as a policy implication, we highlight the importance of understanding the individual needs and willingness of companies when providing government subsidies for enhancing resilience. Furthermore, subsidies should not be limited to only one or two types of SCRESTs, because the types of SCRESTs required vary by industry sector. This highlights the importance of conducting further empirical research.

## 7. Conclusions

Enhancing the resilience of global logistics and supply chains has recently attracted considerable interest from researchers, as its importance was highlighted by the global and widespread impact of the COVID-19 pandemic. In this study, we investigated the impacts of the COVID-19 pandemic on logistics and supply chain activities and the financial performance of Japanese companies, and examined the status of resilience preparedness, response, and future plans from the logistics and supply chain perspective. Finally, we determined whether the existing SCRESTs (if implemented by the interviewed companies) helped them withstand, respond to, or recover from the impacts of the COVID-19 pandemic.

The interview results indicated that not all companies have invested in enhancing logistics and supply chain resilience and that the level of preparedness depends on the company size and the industry sector. Only approximately half of the interviewed companies had plans to enhance logistics and supply chain resilience, which is sensible because companies are not legally bound to compensate for damages and delays caused by natural

disasters and pandemics. However, we argue that companies can benefit from implementing strategic SCRESTs to avoid, withstand, respond to, or recover from the impacts of future disruptions and minimise financial losses. Qualitative interviews allowed us to capture the respondents' subjective experiences of the pandemic. The interviews added vividness, concreteness, and richness to the study.

It is difficult to determine whether popular SCRESTs such as redundant inventory or production capacity, diversification of suppliers, reshoring, and nearshoring satisfy the individual needs of the companies examined in this study. Generally, the cost of maintaining a large inventory or spare production capacity outweighs the gains from mitigating risks—particularly in the case of low-probability events. However, further research is needed on this topic. In the aftermath of the pandemic, the Japanese government provided earmarked subsidies for SCRESTs such as onshoring and nearshoring. Our findings revealed that these SCRESTs were not suitable for the interviewed companies.

The main limitation of this study is its small sample size. Consequently, it is difficult to derive practical implications. Future research should focus on conducting more interviews and large-scale surveys with companies in different sectors to achieve the research objectives.

**Author Contributions:** Conceptualisation, R.M. and H.K.; methodology, R.M.; formal analysis, R.M.; investigation, R.M.; resources, R.M.; writing—original draft preparation, R.M.; writing—review and editing, R.M.; supervision, H.K. All authors have read and agreed to the published version of the manuscript.

**Funding:** This study obtained financial support from the Nippon Foundation.

**Informed Consent Statement:** Informed consent was obtained from all subjects involved in the study.

**Data Availability Statement:** The data presented in this study are contained within the article. The semi-structured questionnaire used in this study is available on request from the corresponding author. The questionnaire used is not publicly available due to the questions being in Japanese language.

**Acknowledgments:** The authors thank the interviewees for sharing valuable information and knowledge. They also like to thank the anonymous reviewers for their comments and suggestions and Nippon foundation for the financial support.

**Conflicts of Interest:** The authors declare no conflict of interest.

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
