# Peer review of "Logistics and Supply Chain Resilience of Japanese Companies: Perspectives from Impacts of the COVID-19 Pandemic"

_logistics, 2023_

Round 1

Reviewer 1 Report

Dear authors, this paper has an interesting scope, but I recommend you some suggestions to improve the quality of this paper:

1º. Authors do not follow the journal's rules from abstract to references sections.

2º. The title is very long. Indeed, I would remove  Global logistics when you are analysing Japanese companies. 

3º. The abstract section must shows Objectives, Methods, findings...Authors need to displays the most relevant information in this section.

4º. Authors have to add updated studies, please. We are researchers and we are obliged to present the last information in this topic. Do not forget!!! We are in 2023. Studies conducted from 2000 to 2015 can be very old in this research, logistics and air transport industries are conditioned  by new technologies. I recommend authors implement these updated studies in the introduction section: 

Florido-Benítez, L. (2023). The Role of the Top 50 US Cargo Airports and 25 Air Cargo Airlines in the Logistics of E-Commerce Companies. Logistics, 7(1):8. https:// www.mdpi.com/2305-6290/7/1/8 

Florido-Benítez, L. (2022). The Safety-Hygiene Air Corridor between UK and Spain will coexist with COVID-19. Logistics, 6(52), pp.1-22. https://doi.org/10.3390/logistics6030052 

da Silva RM, Frederico GF, Garza-Reyes JA. Logistics Service Providers and Industry 4.0: A Systematic Literature Review. Logistics. 2023; 7(1):11. https://doi.org/10.3390/logistics7010011

van Hoek R, Lebigot D. Design of a Company-Engagement Model for Procurement and Supply Management Classes. Logistics. 2023; 7(1):7. https://doi.org/10.3390/logistics7010007

5º. Authors should add "maritime transportation" as a keyword, due to Japan country is an Island. Moreover, I recommend authors to implement two subsections which include "Maritime transportation", and "Air transportation" at literature review section, supported by a location map to stage the importance of logistics and transportation in the Japan island to companies and customers. Please, implement updated studies in this section to compare the results from other studies. Furthermore, authors speak about Japanese companies, but I did not see real examples of Japanese companies in the literature review section. WHY? It is very important to researchers, managers of companies, and logistics operators.

6º. The introduction section is very long, and  the literature review section does not shows the most important keywords of this paper,  as I mentioned previously.

7º Authors need to explain why you worked in this research, objectives and the gaps of this topic. For instance, To fill this gap, the main challenge of ...... Furthermore, authors should display research questions so:

1º Research question or objectives.

2º Research question or objectives.

8º Add new images or figures which provide new information, please. This paper must shows new contributions to the literature review.

9º. Regarding to the methodology section, authors must explain why you used this method, and this need to be supported by other studies to compare their variables, results, and conclusions. Indeed, authors must include a location map where this research was done, please. Readers must know this relevant information. 

10º Authors say: Specifically, the interview questions were broadly classified as follows: 

1. What is the impact of the COVID-19 pandemic on logistics and supply chain activi ties? 

2. What is the impact of the COVID-19 pandemic on the firm’s financial performance?

3. What is the status (past, present, and future) of logistics and supply chain resilience?

4. Did existing logistics and supply chain resilience measures help 

These questions are focused on only Japanese companies, for this reason, I told you that you need to remove in the title "Global logistics", because you are focused on Japanese companies. In addition, from my point of view, five companies interviewed and analysed are few companies to tackle future results and conclusions. Moreover, these four questions are very general, authors need to face new questions and specific questions which address contingency plans, new protocols and tools, new operational procedures to encourage logistics industry in Japan and other countries. Indeed, in table 4, only shows that the company 1 has planned some measures to response future crisis, the rest of companies have not planned or implemented future plans.

11º It is clear that this research is a case of study, focused on 5 companies.

12º. This paper adds nothing new to the literature review in logistics topic.

13º. I would like to see data and information from these 5 companies to analyse the real impact of COVID-19 in these companies. I did not see this in the paper.

14º. In conclusion section, authors say: "The results of the interviews highlighted that not all companies have invested in enhancing logistics and supply chain resilience, and it largely varies by the size of the companies and the nature of their business."  This response is very evident by companies operators. 

15º. Authors say: "It is difficult to conclude whether popular logistics and supply chain resilience strategies, such as redundancy in inventory or production capacity, diversification of suppliers, reshoring, and nearshoring, meet the individual needs of the companies discussed in this study." I agree with authors with five Japanese companies,  information and data bias is very difficult to displays  a good paper.

16º. Authors should include theoretical and managerial implications, limitations and future research subsections to improve the quality of this "case of study".

Author Response

Reply to comments: Reviewer 1

We gratefully thank the reviewer for the meticulous review, valuable comments, and suggestions on our manuscript. We address each comment and revise our manuscript accordingly reply to comments from the reviewers has been reflected in main text wherever possible. Major changes made in the revised version are highlighted in yellow color in the main text.

  1. Authors do not follow the journal's rules from abstract to references sections.

Reply: We thank the reviewer for the comment. We have initially followed the “Free Format Submission” allowed by the logistics journal.  We have now revised the manuscript in the desired format according to the journal guidelines. In the revised version of the manuscript, we have also added the Acknowledgement section.

  1. The title is very long. Indeed, I would remove Global logistics when you are analysing Japanese companies. 

Reply: Following the reviewer’s suggestion we have revised the title of the paper in Lines 2-3.

  1. The abstract section must shows Objectives, Methods, findings...Authors need to displays the most relevant information in this section.

Reply: We have revised the abstract section following journal guidelines in Lines 9-22 in the revised version.

  1. Authors have to add updated studies, please. We are researchers and we are obliged to present the last information in this topic. Do not forget!!! We are in 2023. Studies conducted from 2000 to 2015 can be very old in this research, logistics and air transport industries are conditioned by new technologies. I recommend authors implement these updated studies in the introduction section: 

Florido-Benítez, L. (2023). The Role of the Top 50 US Cargo Airports and 25 Air Cargo Airlines in the Logistics of E-Commerce Companies. Logistics, 7(1):8. https:// www.mdpi.com/2305-6290/7/1/8 

Florido-Benítez, L. (2022). The Safety-Hygiene Air Corridor between UK and Spain will coexist with COVID-19. Logistics, 6(52), pp.1-22. https://doi.org/10.3390/logistics6030052 

da Silva RM, Frederico GF, Garza-Reyes JA. Logistics Service Providers and Industry 4.0: A Systematic Literature Review. Logistics. 2023; 7(1):11. https://doi.org/10.3390/logistics7010011

van Hoek R, Lebigot D. Design of a Company-Engagement Model for Procurement and Supply Management Classes. Logistics. 2023; 7(1):7. https://doi.org/10.3390/logistics7010007

Reply: We thank the reviewer for the suggestion; accordingly, we have added the latest studies relevant to our study in Section 2. We have also fully revised and restructured the manuscript considering the reviewer’s suggestions. We also added new and relevant literature in Lines 123-136.

  1. Authors should add "maritime transportation" as a keyword, due to Japan country is an Island. Moreover, I recommend authors to implement two subsections which include "Maritime transportation", and "Air transportation" at literature review section, supported by a location map to stage the importance of logistics and transportation in the Japan island to companies and customers. Please, implement updated studies in this section to compare the results from other studies. Furthermore, authors speak about Japanese companies, but I did not see real examples of Japanese companies in the literature review section. WHY? It is very important to researchers, managers of companies, and logistics operators.

Reply: We have restructured the literature review section (starting from Page 3) and added a dedicated section reviewing the impacts of the pandemic on Japanese companies from both academic as well as non-academic perspectives. In lines 123-136, we have reviewed studies conducted on Japanese companies. We would like to highlight that there are no studies investigating the impact of the pandemic on the logistics and supply chain activities of Japanese companies.

We have included the results of the surveys conducted in by CBRE and JETRO in lines 107-122 in the literature review section.

Furthermore, we are unable to present an extensive review of academic studies in the literature review section purely due to the lack of studies investigating the impact of the pandemic and the supply chain resilience of Japanese companies.

  1. The introduction section is very long, and the literature review section does not shows the most important keywords of this paper,  as I mentioned previously.

Reply: We have revised and restructured the Introduction section and Literature review section significantly in the revised version of the manuscript including necessary keywords.

  1. Authors need to explain why you worked in this research, objectives and the gaps of this topic. For instance, To fill this gap, the main challenge of ...... Furthermore, authors should display research questions so:

1º Research question or objectives.

2º Research question or objectives.

Reply: We worked on this topic to address the gaps in the existing literature which are highlighted in Section 2.4 (new section added) of the revised version of the manuscript. The objectives of this study are presented in Lines 76-84 which are as follows,

(1) analyse the impacts of the COVID-19 pandemic on logistics and supply chain activities and firms’ financial performance through a case study of five Japanese companies;

(2) identify the current status of resilience preparedness, response, and future plans from the logistics and supply chain perspective in different Japanese companies; and

(3) determine whether the existing logistics and supply chain resilience measures implemented by the interviewed companies helped to avoid, withstand, respond to, or recover from the impacts of the COVID-19 pandemic.

  1. Add new images or figures which provide new information, please. This paper must shows new contributions to the literature review.

 Reply: Following the reviewer’s suggestion, we have added Figure 2 on Page 10 to the revised version of the manuscript.

  1. Regarding to the methodology section, authors must explain why you used this method, and this need to be supported by other studies to compare their variables, results, and conclusions. Indeed, authors must include a location map where this research was done, please. Readers must know this relevant information. 

Reply: Section 3, the method section provides a clear explanation of why we chose the interview-based approach in Lines 177-196 with reference supporting the use of the methodology that has been selected in this study.

  1. Authors say: Specifically, the interview questions were broadly classified as follows: 
  2. What is the impact of the COVID-19 pandemic on logistics and supply chain activi ties? 
  3. What is the impact of the COVID-19 pandemic on the firm’s financial performance?
  4. What is the status (past, present, and future) of logistics and supply chain resilience?
  5. Did existing logistics and supply chain resilience measures help 

These questions are focused on only Japanese companies, for this reason, I told you that you need to remove in the title "Global logistics", because you are focused on Japanese companies. In addition, from my point of view, five companies interviewed and analysed are few companies to tackle future results and conclusions. Moreover, these four questions are very general, authors need to face new questions and specific questions which address contingency plans, new protocols and tools, new operational procedures to encourage logistics industry in Japan and other countries. Indeed, in table 4, only shows that the company 1 has planned some measures to response future crisis, the rest of companies have not planned or implemented future plans.

Reply: Following the reviewer’s suggestion, we have removed the term global from the title in the revised version of the manuscript.

We agree that five companies are a small sample, but we are currently preparing for conducting a large-scale survey of Japanese companies. Our future work will investigate specific details about contingency plans, new protocols and tools, barriers to the implementation of resilience strategies in the supply chain and how these issues can be solved to make supply chains more resilient.

  1. It is clear that this research is a case of study, focused on 5 companies.

Reply: Yes, this is a case study focused on five companies.

  1. This paper adds nothing new to the literature review in logistics topic.

Reply: This study investigated the implementation of logistics and supply chain-related resilience strategies, the impacts of the pandemic and whether companies observed any benefits from the past implementation of resilience strategies. In the literature review part, we have summarized the impacts of the pandemic on Japanese companies especially focusing on their logistics and supply chain activities, implementation of resilience strategies and benefits associated with its implementation. From the literature review, we were able to identify the gaps summarized in Section 2.4 Lines 161-175 which is a new contribution from the point of view of novelty in the literature review.

  1. I would like to see data and information from these 5 companies to analyse the real impact of COVID-19 in these companies. I did not see this in the paper.

Reply: We tried our best to collect real quantitative data but to no luck. In almost all of the cases when we asked about the quantitative data, the respondents stated that they do not have quantitative data at hand at that time.

  1. In conclusion section, authors say: "The results of the interviews highlighted that not all companies have invested in enhancing logistics and supply chain resilience, and it largely varies by the size of the companies and the nature of their business."  This response is very evident by companies operators. 

Reply: We greatly thank the reviewer for this comment.

  1. Authors say: "It is difficult to conclude whether popular logistics and supply chain resilience strategies, such as redundancy in inventory or production capacity, diversification of suppliers, reshoring, and nearshoring, meet the individual needs of the companies discussed in this study." I agree with authors with five Japanese companies,  information and data bias is very difficult to displays a good paper.

Reply: We greatly thank the reviewer for this comment.

  1. Authors should include theoretical and managerial implications, limitations and future research subsections to improve the quality of this "case of study".

Reply: We greatly thank the reviewer for this comment.

We have added policy implications relating to the results of this study in Section 6, Lines 378-382. The limitation of this study and future research agenda is included in Lines 411-414.

Reviewer 2 Report

It is suggested to increase the analysis of interview results.

Is it more reasonable to use a bar chart or a line chart to represent the negative impact?

It is suggested to add the content of policy recommendations.

Author Response

Reply to comments: Reviewer 2

We gratefully thank the reviewer for the comments and suggestions on our manuscript. We address each comment and revise our manuscript accordingly.

  1. It is suggested to increase the analysis of interview results.

Reply: We greatly thank the reviewer for this comment, accordingly, we have added two new sections. Section 5 and Section 6 on pages 9 and 10 presents the discussion and policy implications analysing the interview results. We added Section 6 on Policy implication to bring the interview results into the context of national policy in the revised version of the manuscript. 

  1. Is it more reasonable to use a bar chart or a line chart to represent the negative impact?

Reply: We greatly thank the reviewer for this comment. As we do not have quantitative data, we could not present the impacts using a bar or line chart.

  1. It is suggested to add the content of policy recommendations.

Reply: We greatly thank the reviewer for this comment. We have added policy implications relating to the results of this study in Section 6, Lines 418-422.

Reviewer 3 Report

I think that to improve the paper authors have to do following things: 

1. modify the part “Literature Review” - add materials about Supply chain resiliency and the COVID-19 impact on the Supply Chains

2. The research methodology - Detailed discussion of the study is proposed - sample selection, study time, test conditions, respondents... Try to summarize the main steps of your methodology before.

3. Add part "Analysis"  - in this part there should be more specific, crucial and essential comments. Add detailed research conclusions.

4. Conclusion - please make sure your conclusions' section undersc ore the scientific value added of your paper. Conclusion should include the answer to the research question posed and further directions of research. The Conclusion should be at the most general level possible. 

5. References - The cited literature is not contains numerous references of the current publication (last 2 years). Add new bibliography - and quote it.

The defect of the work is too small number of organizations surveyed. The study is unrepresentative.

What are the limitations of this study?

Author Response

Reply to comments: Reviewer 3

We gratefully thank the reviewer for the comments and suggestions on our manuscript. We address each comment and revise our manuscript accordingly.

  1. Modify the part “Literature Review” - add materials about Supply chain resiliency and the COVID-19 impact on the Supply Chains

Reply: We thank the reviewer for the suggestion; accordingly, we have revised and restructured the literature review in Section 2. We have added dedicated sections reviewing the impacts of the pandemic on Japanese companies from a logistics and supply chain perspective, the status of implementation of logistics and supply chain resilience strategies, and the benefits of implementing logistics and supply chain resilience strategies from both academic as well as non-academic perspectives. We have also added the latest studies relevant to our study in Section 2.

  1. The research methodology - Detailed discussion of the study is proposed - sample selection, study time, test conditions, respondents... Try to summarize the main steps of your methodology before.

Reply: The details of how this study was conducted, including how the samples were selected, the reason for choosing interview as the data collection method and its suitability, and the reason for employing a semi-structured questionnaire are provided in Lines 177-196.

  1. Add part "Analysis"  - in this part there should be more specific, crucial and essential comments. Add detailed research conclusions.

Reply: In the revised version of the manuscript, we have added Sections 5 and 6 which analyzes and discusses the results of the interview and provides policy implications in relation to government initiatives. Detailed conclusions including limitations, and scope for future study have been revised and are presented in Section 7.

  1. Conclusion - please make sure your conclusions' section underscore the scientific value added of your paper. Conclusion should include the answer to the research question posed and further directions of research. The Conclusion should be at the most general level possible. 

Reply: We thank the reviewer for the suggestion. Accordingly, we have presented detailed conclusions limitations, and scope for future study in Section 7.

  1. References - The cited literature is not contains numerous references of the current publication (last 2 years). Add new bibliography - and quote it.

The defect of the work is too small number of organizations surveyed. The study is unrepresentative.

What are the limitations of this study?

Reply: We have added new bibliography such as the following in the revised version of the manuscript.

Zhang, H. The Impact of COVID-19 on Global Production Networks: Evidence from Japanese Multinational Firms. ERIA Discussion Paper Series No. 364, 2021.

Kiers, J.; Seinhorst, J.; Zwanenburg, M.; Stek, K. Which strategies and corresponding competences are needed to improve supply chain resilience: A COVID-19 based review. Logistics, 2022, 6(1), 12.

The manuscript has also been revised to follow the format requirements of this journal. Accordingly, the ACS style of referencing has been adopted in the revised version.

We agree that we have a very small sample which is the biggest limitation of this study. The limitation of this study has been highlighted in Section 7 Lines 411-414.

Reviewer 4 Report

The reviewed article presents the original results of the research carried out by the authors. It focuses on the issue of resilience of global logistics and supply chains, using Japan as a case study. The topic addressed can be considered highly topical. Also, the chosen research approach can be regarded as appropriate. A partial shortcoming is the limited scope of the conducted questionnaire surveys, limited to conducting interviews in five companies. Considering the title itself, thematically focused on global logistics and distribution chains, it would be appropriate to complement or link the Japanese approach with the insight of e.g. foreign firms, similarly focused. At the same time, interviewing only five companies can be seen as more of an initial validation of the relevance of the questions presented. The research could be expanded to include a more significant number of respondents, as well as the inclusion of companies in the questionnaire survey.

For this reason, I submit to the authors for considering whether to expand the number of respondents, including consideration of possible comparison with entities located outside Japan.

Overall, the article is interesting, but let me recommend the sample size.

Thank you. 

Author Response

Reply to comments: Reviewer 4

We gratefully thank the reviewer for the comments and suggestions on our manuscript. We address each comment and revise our manuscript accordingly.

  1. The reviewed article presents the original results of the research carried out by the authors. It focuses on the issue of resilience of global logistics and supply chains, using Japan as a case study. The topic addressed can be considered highly topical. Also, the chosen research approach can be regarded as appropriate. A partial shortcoming is the limited scope of the conducted questionnaire surveys, limited to conducting interviews in five companies. Considering the title itself, thematically focused on global logistics and distribution chains, it would be appropriate to complement or link the Japanese approach with the insight of e.g. foreign firms, similarly focused. At the same time, interviewing only five companies can be seen as more of an initial validation of the relevance of the questions presented. The research could be expanded to include a more significant number of respondents, as well as the inclusion of companies in the questionnaire survey.

For this reason, I submit to the authors for considering whether to expand the number of respondents, including consideration of possible comparison with entities located outside Japan.

Overall, the article is interesting, but let me recommend the sample size.hank you. 

Reply: We greatly thank the reviewer for this valuable comment. We agree that we have a very small sample which is the biggest limitation of this study. The limitation of this study has been highlighted in Section 7 Lines 411-414. Furthermore, we have limited the scope of this study to highlight the case of Japanese companies at this moment hence we have also revised the title of our study in the revised version of the manuscript.

As the reviewer correctly identified this study is more of an initial validation of the relevance of the questions presented. So as to conduct further and detailed analysis, as a part of our future study we are currently preparing to conduct a large-scale survey to identify the details of the impacts of the pandemic on Japanese companies in different industry sectors from a logistics and supply chain perspective, the status of implementation of logistics and supply chain resilience strategies, and the benefits of implementing logistics and supply chain resilience strategies.

Round 2

Reviewer 1 Report

Dear authors, the paper has considerably improved, but authors did not add updated studies like I recommend you.

Florido-Benítez, L. (2022).The World airport awards as a quality distinctive and marketing tool for airports. Journal of Airline Operations and Aviation Management, Vol. 1, No. 2, pp. 54–81. https://doi.org/10.56801/jaoam.v1i2.4

Author Response

Reply to comments: Reviewer 1

Dear authors, the paper has considerably improved, but authors did not add updated studies like I recommend you.

Florido-Benítez, L. (2022).The World airport awards as a quality distinctive and marketing tool for airports. Journal of Airline Operations and Aviation Management, Vol. 1, No. 2, pp. 54–81. https://doi.org/10.56801/jaoam.v1i2.4

Reply: We thank the reviewer for the suggestion; accordingly, we have added the latest studies relevant to our study in Section 2. However, we did not include the suggested paper because it does not fit within the scope or context of our study.